# Lessons Learned from the Policies Developed for the Management of the COVID-19 Pandemic in Northern Cyprus: A Mixed-Methods Study

**DOI:** 10.3390/healthcare13192475

**Published:** 2025-09-29

**Authors:** Seren Fatma Osmanogullari, Nazemin Gilanliogullari, Macide Artac Ozdal

**Affiliations:** 1Department of Health Management, Institute of Graduate Studies & Research, European University of Lefke, TR-10 Mersin, Lefke 99770, Northern Cyprus, Turkey; mozdal@eul.edu.tr; 2Department of Physiotherapy and Rehabilitation, Faculty of Health Sciences, European University of Lefke, TR-10 Mersin, Lefke 99770, Northern Cyprus, Turkey; ngilanliogullari@eul.edu.tr

**Keywords:** COVID-19, health policy, pandemic preparedness, policy analysis, small island states

## Abstract

**Background/Objectives**: The COVID-19 (Coronavirus Disease, 2019) pandemic affected all countries in a variety of ways, and forced policymakers to adapt national health infrastructure. In this context, the strategic adaptation and policy evolution of small island states are understudied. Therefore, the objective of this study was to quantitatively analyse the relationship between confirmed COVID-19 cases and health policy decisions in Northern Cyprus. We also examined the shifting management strategies employed during the pandemic using a replicable statistical analysis framework. **Methods**: In this mixed-methods study, we used systematic thematic analysis to categorise official policy decisions from March 2020 to December 2022. Yearly linear regression models using SPSS and Python correlated the monthly number of decisions with the number of confirmed COVID-19 cases. The analyses included *R*^2^ values, *p*-values, and visualisations with 95% confidence intervals. **Results**: The findings of this study highlight a three-phase strategic period. In 2020, the results (*R*^2^ = 0.03, *p* = 0.63) showed no significant relationship, indicating initial uncertainty. The results (*R*^2^ = 0.60, *p* = 0.003) indicate a strong negative correlation in 2021, which reflects the consistency of the proactive suppression strategies adopted. Conversely, for 2022, the results (*R*^2^ = 0.79, *p* < 0.001) show a strong positive correlation representing the shift to a reactive mitigation strategy, in which the government responded based on case peaks. **Conclusions**: This study’s primary finding is that strategic agility was key to managing the pandemic. For small island states in particular, the effectiveness of geographic advantages like border control depends on a coherent strategy that transcends initial uncertainty. Our data-driven framework provides a tool for analysing this strategic evolution and guiding responses to future pandemics.

## 1. Introduction

A pandemic is the spread of a new disease that affects many people across multiple countries or continents [1]. The world has suffered from viral outbreaks in the last twenty years, such as Coronavirus Disease 2019 (COVID-19), which has resulted in severe mortality, morbidity, and global public health crises [2,3].

COVID-19, which is caused by the coronavirus SARS-CoV-2 [4], was designated a pandemic by the World Health Organisation (WHO) in March 2020. This pandemic negatively impacted global health [5], with 775,431,269 COVID-19 cases and 7,047,741 deaths reported from March 2020 to May 2024 globally. However, these numbers are estimated to be higher [6,7]. The COVID-19 pandemic demonstrated the need to strengthen health systems and policies worldwide to effectively prevent, prepare, detect, and recover from public health threats [8].

The COVID-19 pandemic emerged as a multi-layered public health challenge that profoundly impacted the vulnerability of health systems, social structures, and individual mental health. It is more than just a virological crisis [9]. In this complex context, governments faced challenges in establishing the effectiveness of basic measures like masks and social distancing while addressing secondary issues created by these interventions. The timing and effectiveness of strict lockdowns and border control sparked intense global debates [10].

While these debates continued globally, initial evidence, particularly from island states like New Zealand and Taiwan, demonstrated the potential effectiveness of interventions such as decisive and early implementation of border control in preventing the spread of the virus [11,12].

A study of COVID-19 policies in Australia, Singapore, New Zealand, Finland, and Iceland found that quarantine measures, travel restrictions, and monitoring and surveillance practices led to better pandemic management [13]. Another study in the Philippines found that strict border control, early implementation of lockdowns, and establishment of quarantine facilities effectively reduced and controlled the number of cases [14]. Another study demonstrated that implementing quarantine, contact tracing, screening, and isolation measures significantly mitigated the spread of COVID-19. Quarantine should be implemented early to improve outcomes for larger groups of people. Moreover, controlling travel can enhance the effectiveness of quarantine processes [15].

These initial findings highlighted the critical need for more evidence-based case studies to understand policy implications across diverse geographic and administrative contexts. To contribute to and address this global policy debate, this study examines the case of Northern Cyprus as a “natural laboratory”. Northern Cyprus is a small island state with a centralised administrative structure and controllable borders, allowing for a more precise analysis of the relationship between policy interventions and epidemic dynamics that is free from the confounding factors found in studies of larger countries.

The first positive case of COVID-19 in Northern Cyprus was documented in a German tourist on 10 March 2020 [16]. The Northern Cyprus Ministry of Health Surveillance Report documented 120,692 positive cases from March 2020 to May 2023 [16]. Northern Cyprus suffered from resource limitations, including healthcare infrastructure and testing capacity. The Ministry of Health implemented methods to prevent the spread of the pandemic countrywide, including closing the borders, mandating quarantine for travellers, and enacting public health policies [17].

This study aims to statistically model the three-year policy evolution of Northern Cyprus, providing an evidence-based and concrete case study for the global debate. The main objective is to uncover the distinct statistical signatures of different strategic phases and translate this data-driven narrative into concrete lessons that will provide evidence-based guidance for future pandemic preparedness.

## 2. Materials and Methods

This study used a mixed-methods research design, which is an approach that combines quantitative and qualitative methods into a single study to provide a more comprehensive understanding of a problem [18]. Mixed methods were used to assess the relationship between trends in confirmed COVID-19 cases and health policy decisions in Northern Cyprus. The analysis period covered March 2020 to December 2022. The collected data were used only for this study and were not shared with anyone. The European University of Lefke’s Ethics Committee approved this study on 14 January 2022 (approval number: BAYEK003.08).

### 2.1. Data Collection

Qualitative Data: The data analysed included all official decisions related to the COVID-19 pandemic that were published in the Official Gazette of Northern Cyprus within the period studied. This criterion excludes ministerial guidelines, press announcements, or local advisories to maintain a consistent and high standard for the data corpus. All decisions were systematically reviewed and filtered using the keyword “COVID-19”, and only health-related decisions were chosen for clarity. In this process, a total of 987 health-related policy decisions were inspected and collected. The collected decisions and their published date and numbers were entered into a Microsoft Excel spreadsheet for each year.

Quantitative Data: Secondary data on the number of COVID-19 patients from March 2020 to December 2022 were obtained from the official website of the Northern Cyprus Ministry of Health. Furthermore, daily data were aggregated into monthly totals for each year using a Microsoft Excel spreadsheet for further analysis.

### 2.2. Qualitative Data Analysis

We followed a six-phase thematic analysis approach, as outlined by Braun and Clarke, to perform qualitative analysis of the collected health policy decision data. Thematic analysis is a qualitative research method that identifies themes in data by examining the subject holistically during analysis and reporting. It can be used to understand the research topic and its dimensions [19]. Therefore, the systematic framework adopted in this study ensured transparency in the development of the themes. In the first stage of thematic analysis, we read all health policy decisions multiple times to familiarise ourselves with the data. In stage two, initial codes were systematically generated from the data. In stage three, these initial codes were combined to form the potential themes. After that, the coded excerpts and the overall dataset were carefully reviewed, and the potential themes were refined in stage four. The final themes were identified and named in stage five, and in stage six, a final report that contained the analysis was generated.

An inter-rater reliability approach was integrated into the coding process to ensure the findings were trustworthy. Initially, two researchers (Authors 1 and 2) randomly selected a 20% sample of the health policy decisions and coded them independently. The researchers then came together, compared their coding, discussed any disagreements, and reached a consensus. This process was then used to refine the final version of the coding scheme and applied to the entire dataset consistently. We considered thematic saturation to have been achieved when there were no new codes or themes developed from analysis of the data. For transparency regarding the analytical process, Table 1 presents concrete examples of how raw text from health policy decisions was meticulously coded and subsequently assigned to a final theme.

### 2.3. Quantitative Data Analysis

To first identify and then model the relationship between the monthly volume of health policy decisions (independent variable) and confirmed COVID-19 case numbers (dependent variable), quantitative analysis was conducted for each year of the study period (2020, 2021, and 2022) in sequence. Pearson’s correlation coefficient was selected for the initial test because it is conventional and the most straightforward approach for evaluating a linear association between two continuous variables, which effectively addressed our main research question. At this stage, a dual-method approach was also applied to ensure the accuracy of the findings.

The primary statistical analysis was conducted to assess the strength and direction of the linear relationship between the two variables using Pearson’s correlation coefficient (r) [20] in IBM SPSS Statistics for Windows, Version 31.0. Subsequently, a linear regression analysis was performed to model the association. The coefficient of determination (*R*^2^) and *p*-value were calculated to evaluate the magnitude and statistical significance of the results, with a significance threshold set at *p* < 0.05.

All analyses were replicated using the Python programming language (Version 3.9) to ensure the robustness of the findings. At this point, there were two key aims: first, to cross-validate the statistical results obtained from SPSS, and second, to generate advanced data visualisations, which are presented in this article. The final data visualisations, which included scatter plots with linear regression trend lines and 95% confidence intervals, were created using Python’s Seaborn and Matplotlib (Version 3.9) libraries (Refer to Appendix A). This dual-software approach allowed us to benefit from the established reliability of SPSS for core statistical computation and the superior flexibility and transparency of Python for data visualisation and result confirmation.

## 3. Results

After analysing health policy decisions and confirmed COVID-19 case data, the findings of this study reveal a clear strategic evolution in pandemic management in Northern Cyprus. The findings are presented in two parts: the first communicates the findings of the qualitative thematic analysis of health policy decisions. Then, the second presents the results of the quantitative regression analysis modelling the relationship between the health policy numbers and confirmed COVID-19 case numbers.

### 3.1. Qualitative Finding

After systematic thematic analysis of 987 health policy decisions, twelve distinct themes were developed that represent the main areas of governmental intervention. A summary of the developed themes, the frequency of decisions, the number of generated codes within each theme, and the total number of confirmed COVID-19 cases per year (2020, 2021, and 2022) are provided in Table 2. After the analysis, “Contagion Precautions”, “Border Controls”, “Work Order”, and “Social Activities” were seen as the prominent themes. Although there were significant variations in their intensity in each year, they reflect the shifting priorities of the public health response.

### 3.2. Quantitative Findings

To quantitatively assess the insights obtained from the qualitative analysis and statistically illustrate the evolution of the pandemic management strategy, the correlation between monthly health policy decisions and confirmed COVID-19 case numbers was analysed for each year. Table 3 presents the key findings of this analysis, providing numerical evidence of the significant strategic shift that occurred over the years. The findings shown in this table are visually presented in the following regression plots generated in Python.

#### 3.2.1. 2020: Absence of a Statistically Significant Relationship

The results showed that there was no statistically significant linear relationship between the total number of monthly health policy decisions and the number of confirmed COVID-19 cases (*R*^2^ = 0.0304, *p* = 0.63) in 2020. As a result of this statistical uncertainty, there was a period of reactive and experimental policy-making as demonstrated in Figure 1. However, the absence of a clear statistical trend does not indicate that there was a lack of action. Conversely, the first year of the pandemic was characterised by some of the most drastic interventions. Although it was statistically chaotic, the government of Northern Cyprus implemented strong policies such as “The complete closure of all air and sea ports” and “full lockdown”, leveraging its geographical advantage as an island state to take control of the spread of the virus in the early stages.

#### 3.2.2. 2021: A Strong Negative Correlation Indicating Proactive Control

A robust, statistically meaningful negative correlation (*R*^2^ = 0.6012, *p* = 0.003), suggesting a proactive suppression strategy, was observed in 2021. The months with a higher volume of health policy decisions were significantly associated with a lower number of confirmed COVID-19 cases, as shown in Figure 2. Decisive interventions belonging to the themes “Contagion Precautions”, “Border Controls”, “Work Order”, and “Social Activities” would have played a critical role in this statistical trend. In early 2021, a health policy decision mandated “a mandatory 14-day centralised quarantine for all individuals entering the country, regardless of vaccination status”, exemplifying the strict, proactive measures characteristic of this period.

#### 3.2.3. 2022: A Strong Positive Correlation Indicating a Reactive Strategy

Another significant and different relationship with a strong positive correlation (*R*^2^ = 0.7931, *p* < 0.001) was identified for 2022, as shown in Figure 3. This encouraging pattern suggests that the months in which the confirmed number of COVID-19 cases peaked were also the months during which the highest number of health policy decisions were enacted. There is clear evidence of a reactive strategy in this pattern, where policy-making was primarily driven by, rather than pre-empting, surges in confirmed case numbers. For example, “The mandatory use of masks is being reinstated in crowded indoor areas and on public transport” is a health policy legislated during a case surge in 2022, which illustrates a move away from broad, pre-emptive lockdowns towards more targeted, responsive measures.

## 4. Discussion

This study offers a micro-scale reflection of global policy-making dilemmas and learning processes by statistically modelling the three-year course of Northern Cyprus’s pandemic response. Our findings are not merely an isolated case study; they also have broad implications, drawing on other international experiences.

The strategic uncertainty in 2020 revealed by our study (*R*^2^ ≈ 0.03) reflects the chaos experienced globally during the early months of the pandemic. The findings show that lockdowns during this period “flattened” infection peaks but did not reduce the total number of cases [21]. This explains the dynamics behind our failure to find a statistically significant relationship. Despite the drastic measures taken, the overall lack of a coherent strategy resulted in no clear impact.

In contrast, the success of proactive suppression measures in 2021 (*R*^2^ = 0.60) is consistent with the experiences of other island states. Similarly to the Caribbean and Greenland, the strict border controls in Northern Cyprus leveraged its geographical characteristics to limit the spread of the virus [22,23]. A study indicating that border controls in Europe reduced case numbers ranging from 6% to 25% [24] provides concrete evidence for the mechanism behind the strong negative correlation we observed. This situation demonstrates that geographical advantages can be a robust public health tool when combined with a coherent strategy.

The transition to a reactive approach (*R*^2^ = 0.79) that became apparent in 2022 is the most obvious evidence of the pandemic’s complexity. This strategic shift becomes significant when integrated with the experiences of developed nations such as Saudi Arabia. The gradual and rigid NPIs (Non-Pharmaceutical Interventions) implemented in these countries initially successfully brought case numbers under control [25]. However, the devastating economic shock [26] inflicted on small businesses by these interventions and the psychological pressure on society demonstrated that such harsh strategies were unsustainable in the long term. Therefore, our 2022 finding is the statistical signature of this global “fatigue” and the new balance being established between public health and economic prosperity. This situation is similar to the reported “double whammy” (variants and loosening public compliance) phenomenon that explains why countries like Saudi Arabia and the United Kingdom have been unable to abandon NPIs despite vaccination [27].

### 4.1. Policy Implications and Lessons for Future Pandemics

This in-depth synthesis offers several critical lessons for future public health crises. The primary and most essential lesson is the importance of strategic flexibility. Managing a pandemic demands constant adjustment, with no definitive solution. Second, the success of policy measures varies according to the stage of the outbreak. In the initial and intermediate phases, substantial interventions that utilise geographical benefits, such as border controls, are crucial. At the same time, as the outbreak approaches an endemic phase, more cost-effective and sustainable measures, such as mask mandates, should take precedence [28]. Finally, policymakers should not overlook “hidden” risks, such as small-scale gatherings, which play a significant role in the spread of the outbreak, though they are less visible [29].

### 4.2. Strengths and Limitations of This Study

The main strength of this study lies in its modelling of a country’s three-year policy trajectory, which combined both qualitative and quantitative data and synthesised this in-depth dialogue with the international literature. However, we must acknowledge that this study has significant limitations concerning both its internal and external validity.

First, regarding internal validity, we did not include quantitative data on vaccination rates, the prevalence of specific viral variants, or public compliance. The absence of these confounding variables prevents direct causal interpretation and highlights a vital subject for future, more sophisticated modelling. Furthermore, utilising the volume of health policies as an indicator of intervention intensity does not capture the different effects of various types of NPIs.

Second, the transferability of our findings warrants a critical note. Although the strategic principles identified here—such as the need for policy agility—offer valuable, broadly applicable lessons, our specific statistical results are deeply tied to the context of Northern Cyprus. The island’s unique “natural laboratory” setting, which is a strength in the context of this analysis, also means its quantitative outcomes cannot be directly generalised to larger, non-island nations. Therefore, this study should be seen not as a universal blueprint but rather as a replicable analysis framework that other regions can adapt to their own studies.

## 5. Conclusions

In conclusion, this study statistically demonstrates that the pandemic response in Northern Cyprus was not a monolithic strategy but a clear three-phase evolution from uncertainty (2020) to proactive control (2021) and, finally, reactive mitigation (2022). The key actionable lesson is that effective pandemic management is a function of strategic agility. This translates into three clear imperatives for policymakers for future preparedness: first, to develop adaptive and multi-phased response plans, not static rules; second, to build a data infrastructure that guides strategic shifts in real time; third, to prioritise sustainable interventions, for which it is essential to recognise that the effectiveness of drastic measures, such as border controls, is often time limited and must be balanced against long-term socio-economic realities.

## Figures and Tables

**Figure 1 healthcare-13-02475-f001:**
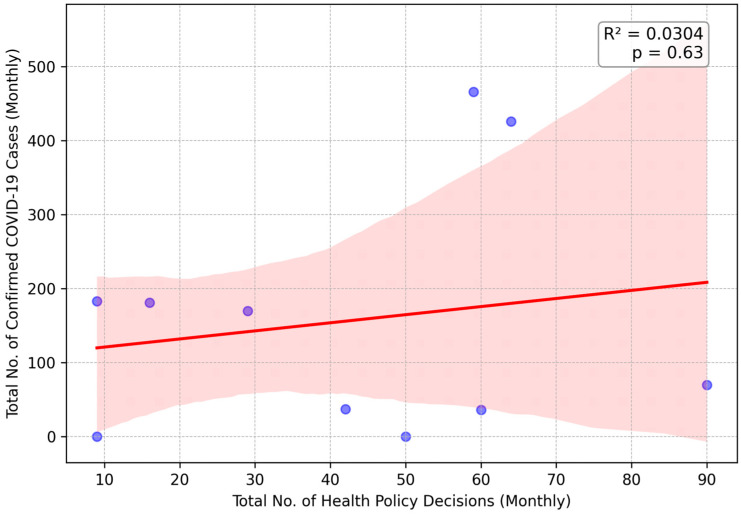
Linear Regression of Monthly Health Policy Decisions on COVID-19 Cases in 2020. Each blue dot represents the data for a single month. The solid red line indicates the linear regression trendline, and the shaded pink area represents the 95% confidence interval for this trend.

**Figure 2 healthcare-13-02475-f002:**
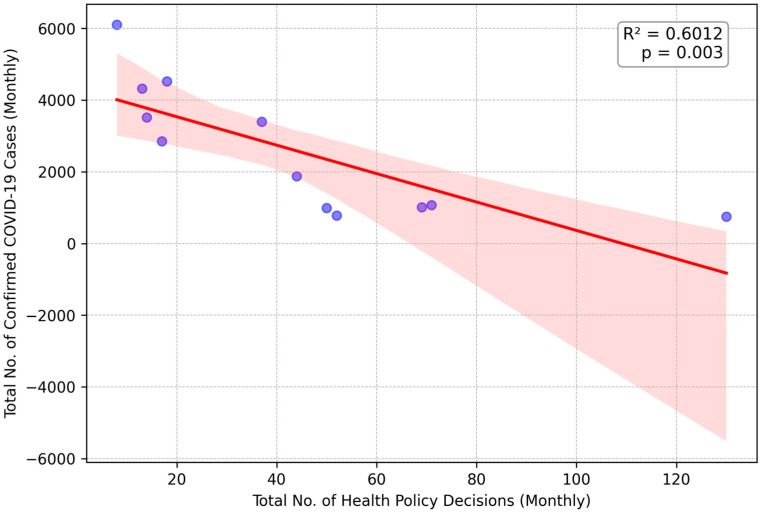
Linear Regression of Monthly Health Policy Decisions on COVID-19 Cases in 2021. Each blue dot represents the data for a single month. The solid red line indicates the linear regression trendline, and the shaded pink area represents the 95% confidence interval for this trend.

**Figure 3 healthcare-13-02475-f003:**
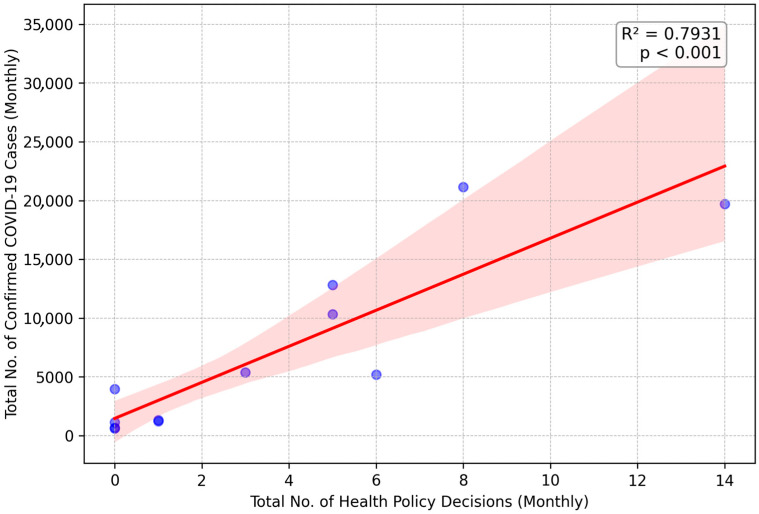
Linear Regression of Monthly Health Policy Decisions on COVID-19 Cases in 2022. Each blue dot represents the data for a single month. The solid red line indicates the linear regression trendline, and the shaded pink area represents the 95% confidence interval for this trend.

**Table 1 healthcare-13-02475-t001:** An example of the coding process.

Date	Official Gazette No.	Decisions	Codes	The Main Theme
26 January 2021	21	Those who engage in contactless trade within the scope of the Green Line regulation can trade without quarantine.	Green Line	Trade
9 April 2020	62	All land, sea, air and entry gates are banned for tourists and all other country citizens except for Northern Cyprus citizens, their spouses and children.	Entry gates	Border Controls
28 January 2021	23	All public and private schools will switch from face-to-face education to online education.	Online education, face to face education	Education
19 March 2021	64	Starting home quarantine, establishing wristbands and tracking systems	Home quarantine	Contagion Precautions
12 July 2021	155	Obligation to comply with mask, distance and 1.5 m rules in all indoor and outdoor areas	mask, distance and 1.5 m rules	
19 February 2021	40	Suspension of takeaway services along with restaurants	Takeaway services	Work Order
4 April 2020	59	Initiating criminal proceedings against those who do not comply with the partial curfew	Criminal proceedings	Penalties and Control
5 July 2020	126	Receiving services from private health institutions for the COVID-19	Private health institutions	Health Services

**Table 2 healthcare-13-02475-t002:** This table represents the main themes, the annual total number of health policy decisions and codes, and the number of confirmed COVID-19 cases for 2020, 2021, and 2022.

Main Themes	Total No. of Decisions			No. of Thematic Codes			Total No. of Confirmed COVID-19 Cases per Year		
	2020	2021	2022	2020	2021	2022	2020	2021	2022
Transportation	7	5	1	2	2	1	1508	31,261	83,517
Tourism	1	8	0	1	2	0			
Basic Needs	2	6	0	1	1	0			
Social Activities	51	53	1	16	14	1			
Border Controls	82	72	8	6	4	2			
Health Services	9	4	1	4	1	1			
Trade	8	7	0	3	3	0			
Education	19	21	3	3	5	3			
Contagion Precautions	151	258	28	15	20	6			
Official Notice and Permission	10	5	0	4	2	0			
Work Order	63	82	0	24	24	0			
Penalties and Controls	20	1	0	7	1	0			

**Table 3 healthcare-13-02475-t003:** Statistical summary of the yearly relationship between heath policy decisions and confirmed COVID-19 cases.

Year	Total No. of Policy Decisions (Yearly)	Total No. of Confirmed COVID-19 Cases (Yearly)	Pearson’s *r* Value	R Square (*R*^2^) Value	*p*-Value	Interpretation of Relationship
2020	423	1508	0.1743	0.030	0.630	Not Statistically Significant
2021	522	31,261	−0.7754	0.601	0.003	Strong and Significantly Negative
2022	42	83,517	0.8906	0.793	0.001	Strong and Significantly Positive

Pearson’s *r* Value: Reflects both the direction and the strength of the association between two variables. A value close to −1 suggests a strong negative (inversely directed) association, whereas a value close to +1 signifies a strong positive (directionally directed) association. R Square (*R*^2^) Value: Indicates the proportion of variability in the number of cases that can be accounted for by the variability in the number of decisions. A high percentage signifies a strong association. *p*-Value: Demonstrates whether the observed relationship is statistically significant. Typically, a *p*-value below 0.05 suggests that the finding is unlikely to be due to chance and is deemed statistically significant.

## Data Availability

The data presented in this study are available upon request from the corresponding author due to ethical reasons.

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
