# Peer review of "Lessons Learned from the Policies Developed for the Management of the COVID-19 Pandemic in Northern Cyprus: A Mixed-Methods Study"

_healthcare, 2025, doi:10.3390/healthcare13192475_

Round 1
Reviewer 1 Report (Previous Reviewer 1)
Comments and Suggestions for Authors
Thank you for the opportunity, and I think the manuscript has been improved well.
And consider the following comment.
Table 3.2. has missing variables: the number of confirmed cases for each year and the number of health policy decisions should be included in the table. These are the two main variables; without them, the table will not have meaning.
Author Response
Comment 1: Thank you for the opportunity, and I think the manuscript has been improved well.
And consider the following comment.
Table 3.2. has missing variables: the number of confirmed cases for each year and the number of health policy decisions should be included in the table. These are the two main variables; without them, the table will not have meaning.
Response 1:
We thank Reviewer 1 for their positive feedback that the manuscript has been “improved well” and for their crucial suggestion.
We have now added two new columns to Table 3.2: “Total No. of Policy Decisions (Yearly)” and “Total No. of Confirmed COVID-19 Cases (Yearly)”. This ensures that the table is now self-contained, presenting both the raw data totals and the statistical results of the relationship between them, making it far more meaningful for the reader.
Reviewer 2 Report (Previous Reviewer 3)
Comments and Suggestions for Authors
In the methods section the process of selecting the 774 policy decisions should be clarified: were these only decisions published in the Official Gazette, or were additional policy levels (e.g., local or ministerial guidelines) considered? Moreover, the dual use of SPSS and Python is interesting, but the justification should be more concise, highlighting that Python was mainly employed for cross-validation and advanced visualisation.
The discussion is now generally well developed, but at times descriptive rather than critical. It could be strengthened by expanding on the limitations of the study (e.g., absence of vaccination rates, variant emergence, public compliance, human mobility data); more critically discussing the transferability of the findings: to what extent are the lessons from Northern Cyprus generalisable to other small states or larger nations? Offering more concrete, practice-oriented recommendations for policymakers.
Likewise, the conclusion section effectively summarises the trajectory of policy strategies. However, it could be made more concise and direct, emphasising the “lessons learned” in clear and actionable terms rather than repeating earlier narrative detail.
Finally, the manuscript would benefit from careful language polishing. Some sentences, especially in the Introduction and Discussion, are overly long and complex, which affects readability.
In conclusion, I think that with these adjustments, the article will be considerably strengthened and will provide a robust case study of policy evolution in pandemic management.
Author Response
We are very grateful to Reviewer 2 for their thoughtful and detailed guidance, which has helped us to sharpen the sections of our paper critically.
Comment 2.1: In the methods section the process of selecting the 774 policy decisions should be clarified: were these only decisions published in the Official Gazette, or were additional policy levels (e.g., local or ministerial guidelines) considered? Moreover, the dual use of SPSS and Python is interesting, but the justification should be more concise, highlighting that Python was mainly employed for cross-validation and advanced visualisation.
Response 2.1:
• We have added a clear statement to the “2.1. Data Collection: Qualitative Data” section to clarify the scope of our data corpus, specifying that our analysis was confined exclusively to the Official Gazette to ensure the use of verifiable and legally binding directives.
• As suggested, we have revised the justification for our dual-software approach to be more concise. The text in the “Quantitative Data Analysis” section now highlights more directly that SPSS was used for primary statistical computation. At the same time, Python was employed for its superior capabilities in cross-validation and advanced data visualisation.
Comment 2.2: The discussion is now generally well-developed, but at times descriptive rather than critical. It could be strengthened by expanding on the limitations of the study (e.g., absence of vaccination rates, variant emergence, public compliance, human mobility data); more critically discussing the transferability of the findings: to what extent are the lessons from Northern Cyprus generalisable to other small states or larger nations? Offering more concrete, practice-oriented recommendations for policymakers. Likewise, the conclusion section effectively summarises the trajectory of policy strategies. However, it could be made more concise and direct, emphasising the “lessons learned” in clear and actionable terms rather than repeating earlier narrative detail.
Response 2.2: We are very grateful for this set of insightful and highly constructive recommendations, which have significantly elevated the critical depth of our manuscript’s final sections. We have addressed each of the points raised:
• Expanding on Limitations and Transferability: We have completely rewritten the “Strengths and Limitations” section to be more comprehensive and critical. As suggested, we now expand on how the absence of confounders (like vaccination rates and variant emergence) impacts the interpretation of our findings. Critically, we have also integrated the discussion on transferability directly into this section. This new and unified paragraph now makes a clear and nuanced distinction between the study’s broadly applicable strategic lessons and its context-dependent statistical outcomes, directly addressing the question of generalisability in a more methodologically sound manner.
• Offering Concrete Recommendations and a More Concise Conclusion: We took this advice to heart by fundamentally reshaping the study’s final message. The recommendations in our “Implications” section have been made more practice-oriented. More importantly, we have completely restructured the Conclusion to be more direct and actionable. It no longer repeats the narrative; instead, it now emphasises the key “lessons learned” as three clear, numbered imperatives for policymakers, delivering the concise and impactful takeaway the reviewer suggested.
We believe that these changes have transformed the Discussion and Conclusion from a descriptive summary into a more critical, practical, and powerful synthesis of our findings.
Comment 2.3: Finally, the manuscript would benefit from careful language polishing. Some sentences, especially in the Introduction and Discussion, are overly long and complex, which affects readability.
Response 2.3: We thank the reviewer for this feedback. We have carefully revised the manuscript to improve its readability, with a particular focus on shortening complex sentences. To further ensure the highest linguistic standards, the paper has also been professionally edited by MDPI's English Language Editing service.
Comment 2.4: In conclusion, I think that with these adjustments, the article will be considerably strengthened and will provide a robust case study of policy evolution in pandemic management.
Response 2.4: We sincerely thank the reviewer for their concluding remark that these adjustments would create a “robust case study”. We hope that we have achieved that goal.
Reviewer 3 Report (Previous Reviewer 4)
Comments and Suggestions for Authors
Thank you for the opportunity to review the manuscript titled “Lessons Learned from the Policies Developed for the Management of the COVID-19 Pandemic in Northern Cyprus: A Mixed Methods Study”. I appreciate the authors’ efforts in addressing the reviewer’ comments. Wishing you success with the manuscript.
Author Response
Comment 1: Thank you for the opportunity to review the manuscript titled “Lessons Learned from the Policies Developed for the Management of the COVID-19 Pandemic in Northern Cyprus: A Mixed Methods Study”. I appreciate the authors’ efforts in addressing the reviewer’s comments. Wishing you success with the manuscript.
Response 1:
We sincerely thank Reviewer 3 for their positive feedback and for their support of our revision efforts. We are very grateful for their encouragement and good wishes.
Round 2
Reviewer 2 Report (Previous Reviewer 3)
Comments and Suggestions for Authors
Thank you to the authors for revised the manuscript.
Now this study, in my opinion, is ready for publication.
Author Response
Comment: Thank you to the authors for revised the manuscript.
Now this study, in my opinion, is ready for publication.
Response:
We are sincerely grateful to Reviewer 2 for their positive assessment and for their final recommendation. Their rigorous and insightful feedback was instrumental in significantly strengthening this manuscript, and we are delighted that they find the revised version ready for publication.
This manuscript is a resubmission of an earlier submission. The following is a list of the peer review reports and author responses from that submission.
Round 1
Reviewer 1 Report
Comments and Suggestions for Authors
Thank you so much for this opportunity to review your paper. Please, see below a few suggestions you might consider or clarify:
Materials and methods
- The first paragraph started with the sentence: “This study uses mixed-methods research……”
Since this is a research report, it should be written with past tense.
- Why do you choose the mixed method?
- You mentioned a retrospective cohort study but I couldn’t find anything related to this particular study design.
- Where is the exposure status (exposed vs non exposed)?
- The outcome is not clearly indicated
- What is the sample size?
- How did you identify exposure status?
- Why you choose to use Microsoft excel while we have a lot of strong and trustworthy other software? Like SPSS, STATA……
- Did you measured a relative risk (RR)?
- How did you control the quality of your data?
- How did you analyze the qualitative part?
- Have you used any qualitative data analysis software?
- For the qualitative part, what was the method of data collection?
Result
- It is almost all in all the findings of the qualitative part.
Discussion
- It is also about qualitative findings.
Reviewer 2 Report
Comments and Suggestions for Authors
I read with interest the study presented by Osmanogullari and colleagues on the policies enacted in Northern Cyprus to manage the COVID-19 pandemic. Two key findings stood out: first, the identification of thematic categories under which 774 pandemic-related decisions were grouped; and second, the apparent correlation between policy enforcement and fluctuations in case numbers across the timeline of the pandemic.
The topic aligns well with Healthcare’s aim and scope, particularly as it offers a country-level case study on health policy response during a public health emergency. The manuscript is of potential interest to the journal’s readership, especially policymakers and public health professionals focused on emergency preparedness and comparative policy analysis.
Despite its merits, the manuscript is riddled with issues that need to be addressed before it can be accepted for publication.
1- While the study's objectives are vaguely stated, a clear rationale guiding the research question is missing. The introduction fails to build a coherent argument on why Northern Cyprus was uniquely suited for this analysis.
2- The results section describes decisions and case numbers but lacks analytical rigor. No statistical testing or modeling was used to assess causality or significance.
3- Although the study claims to be mixed-methods, the qualitative thematic analysis is described superficially and is not integrated with the quantitative findings. This undermines the essence of mixed-methods research.
4- While the manuscript includes figures showing decisions and case numbers, there is little narrative analysis of these visuals. The authors must guide readers through what the figures demonstrate and how they support their arguments. Furthermore, figures 1 and 3 are not present properly and parts of them are missing.
5- The discussion often reiterates the results and compares the findings with previous literature in a general manner without deeper synthesis. For example, references to studies in New Zealand or South Korea are not critically compared to the Northern Cyprus context. This is unacceptable given that many regional studies have attempted to address the same issue in this study, e.g., Saudi Arabia (10.3390/ijerph18020783) and (10.7759/cureus.33042), (10.3390/healthcare11121757).
6- The manuscript implies that decisions caused declines in COVID-19 cases, yet no formal analytical strategy is used to test this. Without controlling for confounders such as seasonality or vaccination coverage, such claims are speculative.
7- The ethics approval is reported, but data sourcing remains vaguely described. There is no discussion of potential bias in manual policy coding or steps taken to ensure reliability and validity in the thematic analysis.
8- The final paragraph makes sweeping recommendations (e.g., adopt more integrated strategies) without grounding these in the findings or acknowledging the study's limitations.
Comments on the Quality of English LanguageSeveral paragraphs suffer from awkward phrasing and grammatical errors. Examples include tautologies ("we should adopt more integrated strategies to enhance...") and misplaced conjunctions ("We, to better prepare...").
Reviewer 3 Report
Comments and Suggestions for Authors
Dear Authors,
The manuscript addresses a timely and relevant topic by evaluating pandemic management policies in Northern Cyprus using a mixed-methods approach. The authors aim to extract lessons from the COVID-19 response and offer guidance for future health emergencies. While the intent and scope of the work are commendable, the manuscript presents several methodological, analytical, and structural weaknesses that should be addressed before publication.
First, the introduction fails to sufficiently and widely contextualize the context of the policies applied at the time of the pandemic and the lessons potentially learned from the errors and evidence of the time. In this sense I suggest to implement this aspect with a greater comparative argument (e.g. https://pubmed.ncbi.nlm.nih.gov/39993410/ ; https://pubmed.ncbi.nlm.nih.gov/40525373/ ; https://pubmed.ncbi.nlm.nih.gov/33561814/ ; https://pubmed.ncbi.nlm.nih.gov/36033738/ ; https://pubmed.ncbi.nlm.nih.gov/39970940/ ; https://pubmed.ncbi.nlm.nih.gov/33624696/ ).
Moreover, while a substantial number of policies are categorized and described, the analysis remains largely descriptive. The manuscript does not provide a rigorous statistical correlation between policy themes and infection trends. I reccomend to integrate inferential statistics to support claims about the effectiveness of specific interventions. Even simple interrupted time-series or correlation analyses would significantly enhance credibility.
I note that the qualitative analysis is not grounded in a robust theoretical framework (e.g., Braun and Clarke’s thematic analysis methodology is not fully applied). There is no clear explanation of inter-rater reliability or how thematic saturation was achieved. Please, include a clearer description of how themes were developed, justified, and validated. Provide a coding tree or an example of how raw data was coded into themes.
Furthermore, there is no stratification by age, sex, vaccination status, or healthcare utilization data. Without controlling for these confounders, the attribution of changes in case numbers to specific policies is weak. It is necessary to acknowledge these limitations more prominently and, if possible, incorporate secondary variables or conduct sensitivity analyses.
The authors repeatedly implies causality between policy implementation and case trends without appropriate analytical justification. Please, rephrase claims to reflect associations, unless supported by more robust statistical testing.
In the manuscript figures are referred to, but they are not critically analyzed or statistically interpreted. Also, graphs are not standardized, and tables are more illustrative than analytical.
I suggest to standardize the format of figures, include confidence intervals or trend lines, and perform statistical comparison between peaks and troughs in incidence relative to policy changes.
Finally, the manuscript would benefit from substantial linguistic editing. Phrases like “positive COVID-19 patients” should be revised (e.g., “confirmed COVID-19 cases”). Additionally, sentence construction is often awkward or repetitive.
I hope that my suggestions will be useful to implement this interesting study.
Kind regards
Reviewer 4 Report
Comments and Suggestions for Authors
Kindly find the attached file.
